# Complete bio-degradation of poly(butylene adipate-co-terephthalate) via engineered cutinases

Yu Yang [1,4], Jian Min [1,4], Ting Xue[1,4], Pengcheng Jiang[1], Xin Liu[1], Rouming Peng[1], Jian-Wen Huang[1], Yingying Qu[1], Xian Li[1], Ning Ma[2], Fang-Chang Tsai [2], Longhai Dai [1], Qi Zhang[3], Yingle Liu [3] ✉, Chun-Chi Chen [1] ✉ & Rey-Ting Guo [1] ✉

Poly(butylene adipate-co-terephthalate) (PBAT), a polyester made of terephthalic acid (TPA), 1,4-butanediol, and adipic acid, is extensively utilized in plastic production and has accumulated globally as environmental waste. Biodegradation is an attractive strategy to manage PBAT, but an effective PBAT-degrading enzyme is required. Here, we demonstrate that cutinases are highly potent enzymes that can completely decompose PBAT films in 48 h. We further show that the engineered cutinases, by applying a double mutation strategy to render a more flexible substrate-binding pocket exhibit higher decomposition rates. Notably, these variants produce TPA as a major end-product, which is beneficial feature for the future recycling economy. The crystal structures of wild type and double mutation of a cutinase from *Thermobifida fusca* in complex with a substrate analogue are also solved, elucidating their substrate-binding modes. These structural and biochemical analyses enable us to propose the mechanism of cutinase-mediated PBAT degradation.

Synthetic polyesters such as polyethylene terephthalate (PET) and poly(butylene adipate-co-terephthalate) (PBAT) are among the most popular plastics in our daily life due to their beneficial properties: low cost, lightweight, and durable[1,2]. It is estimated that the annual production of aliphatic-aromatic co-polyesters exceeded 360 million tons in 2018[3]. PBAT, also known as Ecoflex® (BASF, Germany), is a copolymer composed of flexible (butylene adipate) and rigid (butylene terephthalate) segments with various degrees of polymerization (Fig. 1)[4,5]. PBAT has been widely utilized in food packaging, agricultural, textile, and other industries[4,6]. In agriculture, the mulch film made by PBAT is used to improve soil conditions and enhance crops production[7]. In

contrast with polymers constructed via carbon-carbon bonds, PBAT was first considered as a compostable biopolymer because polyesters are more susceptible to enzymatic degradation, such as by esterases[8]. However, albeit easier to be fragmented in the environment, the natural degradation rate of PBAT is quite slow. Therefore, the wide application of PBAT has resulted in large-scale waste accumulation that has caused severe environmental burden. Thus, developing measures to reduce and recycle PBAT waste is important for its sustainable use[8,9].

Bio-based degradation using renewable biological entities, i.e., enzymes or microorganisms, is an ideal and environmentally benign approach to reduce and recycle plastics. Thus, significant effort has

[1]State Key Laboratory of Biocatalysis and Enzyme Engineering, Hubei Hongshan Laboratory, Hubei Collaborative Innovation Center for Green Transformation of Bio-Resources, Hubei Key Laboratory of Industrial Biotechnology, School of Life Sciences, Hubei University, 430062 Wuhan, People's Republic of China. [2]Hubei Key Laboratory of Polymer Materials, Key Laboratory for the Green Preparation and Application of Functional Materials (Ministry of Education), Hubei Collaborative Innovation Center for Advanced Organic Chemical Materials, School of Materials Science and Engineering, Hubei University, 430062 Wuhan, People's Republic of China. [3]State Key Laboratory of Virology, College of Life Sciences, Wuhan University, 430072 Wuhan, People's Republic of China. [4]These authors contributed equally: Yu Yang, Jian Min, Ting Xue. ✉e-mail: mvlwu@whu.edu.cn; ccckate0722@hubu.edu.cn; guoreyting@hubu.edu.cn

**Fig. 1 | Chemical structures of PBAT and PET and the putative ester hydrolysis products.** Similar to PET, biodegradation of PBAT could also yield TPA.

been made to develop a PBAT biodegradation strategy. Recent studies characterized microbes that degrade PBAT, including *Stenotrophomonas* sp. TGJ1, *Cryptococcus* sp. MTCC 5455, *Bacillus pumilus*, and *Knufia chersonesos*[6,10–15]. These microorganisms secrete an array of polyester hydrolyzing enzymes associated with PBAT decomposition. Several hydrolases have also been isolated and biochemically characterized (Supplementary Table 1). Biundo et al. reported a PBAT-hydrolyzing lipase from *Pelosinus fermentans*[11]. Wallace et al. used proteomic screening to discover an esterase from *Pseudomonas pseudoalcaligenes* that hydrolyzes PBAT at an optimal temperature of 65 °C[12]. Nevertheless, the PBAT decomposition rates of lipases or esterases are generally low. The sluggish decomposition rates may be caused by narrow substrate-binding channels that prefer long fatty acids, not aromatic-containing polymers, as substrates[4,11,16]. In comparison with these esterases, cutinases such as those from *Humicola insolens* and *Thermobifida cellulosilytica* that harbor broader substrate-binding cavities have been reported to show higher PBAT decomposition rates[17]. However, the lack of in-depth investigation into their potency and mechanism of action hampers their applications.

The recent progresses in developing PET biodegradation with cutinases or cutinase-like enzymes can be borrowed to advance PBAT biodegradation[1,18,19]. Cutinase is an esterase that belongs to the α/β-hydrolase superfamily and primarily hydrolyzes cutin from plants[20,21]. Unlike lipases, cutinases lack the active site-covering lid structure such that the substrate-binding cleft is exposed and directly accessible from bulk solution[22–24]. In addition, the substrate-binding site is more expansive in cutinases compared to esterases, benefiting the binding of bulkier macromolecules. A recent study reported that the PET hydrolytic activity of a thermophilic cutinase from the leaf-branch compost metagenome named LCC surpasses other homologous enzymes at optimal conditions[18]. The engineered LCC variants with elevated thermostability can decompose >90% PET at 72 °C in 10 h and show higher hydrolytic activity towards reinforced PET materials[18,25]. Another cutinase from a PET-assimilating bacterium termed *Is*PETase exhibits the highest PET hydrolysis rate at ambient temperature[2]. Structural analyses indicate that *Is*PETase is distinct from canonical cutinases by having a flexible substrate-binding pocket that can expand upon binding to the TPA moiety of PET[26,27]. This feature is attributed to the presence of two *Is*PETase-unique smaller residues located under the TPA-binding tryptophan[19,26]. Replacing the equivalent residues in canonical cutinases (His/Phe) by *Is*PETase-unique smaller residues (Ser/Ile) significantly enhances PET-hydrolytic activity. Thus, the double mutation (DM) strategy has been considered a minimal requirement to generate an authentic PET hydrolytic enzyme[19].

In this work, we investigate the PBAT hydrolysis potency of cutinases and PET-degrading enzymes, and find that cutinases outcompete other reported enzymes. Significantly, engineering cutinases with the DM strategy results in higher activity such that PBAT polymers

can be decomposed to their monomeric constituents. We apply biochemical and structural analyses to investigate the hydrolytic intermediates/products and substrate-binding features of a cutinase from *T. fusca* (*Tf*Cut) to propose its mechanism of action. Taken together, this study illustrates the potential application of engineered cutinase in PBAT biodegradation. These results provide an important basis to guide the development of biodegradation/bio-recycling of PBAT as well as other aliphatic-aromatic co-polyesters.

## Results and discussion
### *Tf*Cut-mediated PBAT hydrolysis
We first investigated the PBAT degradation activity of several cutinases and cutinase-like enzyme that have been found to exhibit PET hydrolytic activity[19]. Notably, a cutinase from *T. fusca* (*Tf*Cut)[20,28,29] is the most effective enzyme, which can "crop" PBAT film into fragments in 4 h with further digestion into small particles in 12 h. After 48 h, PBAT was fully decomposed, leaving no visible granules in solution (Fig. 2a). After 4-h incubation, four major intermediates (assigned compounds **1–4**, Fig. 2b) were detected via HPLC ( >99% of total peak area). These compounds should possess similar chromophores judging by their same maximal absorbance at 241 nm (Supplementary Fig. 2). Compounds **1** and **2** exhibit the same retention time and absorbances as TPA and BTa standard samples, respectively. Thus, their identities are reasonably inferred to the corresponding molecules (Supplementary Fig. 3). Various smaller intermediates may be formed via ester bond cleavage (Supplementary Fig. 4). Mass spectra collected in a negative mode indicate that the *m/z* values of **1–4** are 165.2, 237.2, 365.2, and 385.2, which can be posited as TPA, BTa, ABTa, and TaBTa, respectively (Fig. 2c), as no other possible products exhibit equivalent *m/z* values (Supplementary Table 2). Two minor products (<1% of total peak area), compounds **5** and **6**, were also revealed (Supplementary Fig. 5). Compound **1** accumulated over time and dominated in solution, reaching a maximum within 36–48 h, while the amounts of **3** and **4** increased at the beginning and promptly decreased after 12 h to non-detectable levels (Fig. 2d). The amount of **2** increased in 24 h and then slightly dropped. It is worth noting that compounds **1–4** share similar properties as they all contain a terminal TPA with 1,4-butanediol and adipic acid on the other side (Fig. 2c), suggesting a unique mechanism involved in the *Tf*Cut-catalyzed PBAT degradation, especially at the TPA-containing moiety (*vide infra*). Based on the product yields, we found that cutinases or cutinase-like enzymes, except *Pb*PL, are more effective than lipase or esterase in PBAT decomposition and that more than one order of magnitude of hydrolytic products were produced (Supplementary Table 1)[11,12,30]. Notably, a cutinase from *T. cellulosilytica* (*Tc*Cut) that has been previously reported as a potent PBAT-degrading enzyme was also analyzed[17] and was found to exhibit similar activity to *Tf*Cut (Supplementary Table 1). Altogether, cutinases are more effective PBAT-degrading enzymes compared to other types of esterases.

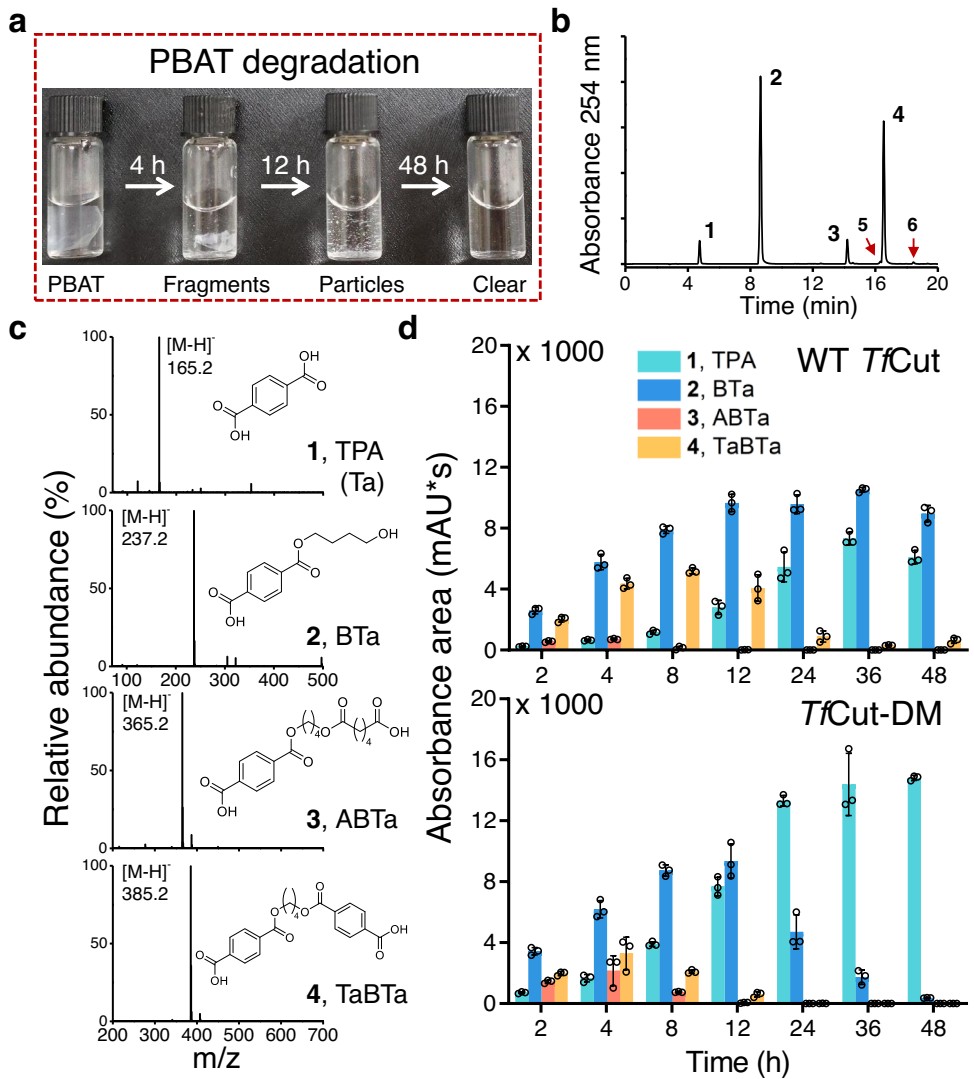

**Fig. 2 | Degradation of PBAT by *Tf*Cut. a** PBAT film hydrolysis catalyzed by *Tf*Cut at 70 °C over time. The optimal temperature of *Tf*Cut was determined and shown in Supplementary Fig. 1. **b** HPLC chromatograms of PBAT hydrolytic products at 4 h. **c** The structures of the intermediates in **b** were identified based on the mass spectrum. **d** The time-course of four major PBAT-hydrolytic intermediates produced by *Tf*Cut treatment. TPA (or Ta), terephthalic acid; B, 1,4-butanediol; A, adipic acid. Triplicate independent assay was performed in each experiment. Data are presented as mean values +/− standard deviation. The individual data points are shown as black circles. Source data are provided as a Source Data file.

## Utilization of the DM strategy to enhance PBAT hydrolytic potency

Our previous study indicates that the DM engineering strategy can improve the PET hydrolytic activity of several enzymes[19]. Thus, we proceeded to investigate the PBAT-degradation activity of DM variants of the abovementioned enzymes. First, *Tf*Cut-DM was analyzed and it produced four major hydrolytic intermediates that share the same HPLC separation profiles as compounds **1**–**4** of WT *Tf*Cut (Supplementary Fig. 3), but no detectable amounts of **5** and **6**, suggesting a similar hydrolytic mechanism is employed. As shown in Fig. 2d, the *Tf*Cut-DM-catalyzed hydrolytic reaction is faster than that of WT *Tf*Cut, such that the complete PBAT decomposition was achieved in 36 h (48 h for WT *Tf*Cut). Notably, *Tf*Cut-DM produced 3-fold more TPA than WT *Tf*Cut within 12 h (Supplementary Table 3). Compound **3** accumulated within 4 h and then decreased to an undetectable level after 12 h in *Tf*Cut-DM-treated samples. Compound **4** has a similar trend, but its highest amount is 1.6-fold less than that in WT *Tf*Cut, which occurs at 8 h. Moreover, TPA accounts for the major product in *Tf*Cut-DM-treated sample (**1**, 97.6%), while there is still a comparable amount of BTa (**2**, 59.5%) in WT *Tf*Cut (Fig. 2d and Supplementary Table 3). Notably, we also found that DM

variants of *Bur*PL and *Tc*Cut exhibit higher PBAT degradation activity than the WT enzymes (Supplementary Fig. 6a), though no significant enhancement was observed for *Pb*PL-DM (data not shown). Intriguingly, *Is*PETase that was engineered to reverse its DM (rDM variant, harboring His/Phe duo) shows lower activity (Supplementary Fig. 6b). Overall, DM variants exhibit higher PBAT-hydrolyzing activity, which is highlighted by their ability to yield higher amounts of TPA as the product.

To mimic conditions enzymes may encounter in the field, we investigated activity with UV-illuminated PBAT films[31]. As revealed in a previous report, the cross-linked polymers in UV-treated films are more resistant to enzyme-mediated hydrolysis and lower amounts of hydrolytic products were detected (Supplementary Fig. 7). Notably, *Tf*Cut-DM is more effective and yields TPA as a main product (Supplementary Fig. 7). These results indicate that the higher potency of DM variants should allow the decomposition of post-consumer PBAT films as well as the original material.

### *Tf*Cut-DM in complex with a substrate analog

In order to understand the mechanism of action of the DM variant of cutinase, we aimed to solve their crystal structures and eventually

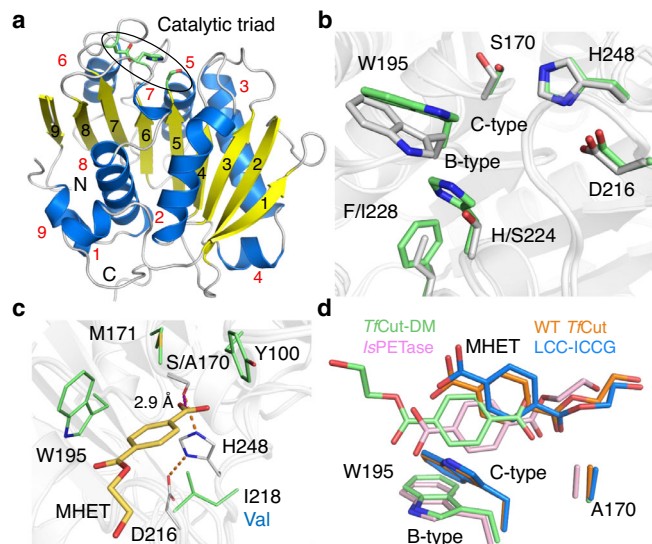

**Fig. 3 | The MHET-bound *Tf*Cut-DM structure. a** The X-ray crystal structure of *Tf*Cut-DM. The numbered α-helices, β-sheets, and loop regions are colored in blue, yellow, and white, respectively. Three amino acids that constitute the catalytic triad are shown in green sticks and circled in black. **b** Superimposed residues in the active site of *Tf*Cut-DM (white, PDB ID: 7XTR) with WT *Tf*Cut (green, PDB ID: 5ZOA) displayed in sticks. **c** Key residues surrounding MHET in the active site are shown as green sticks. The alternative residue in LCC is labeled under I218 in blue color. The catalytic triad residues are shown as white sticks. Residue S170 is from the WT *Tf*Cut (white stick, PDB ID: 5ZOA). **d** Comparison of the substrate-binding poses between MHET-bound WT *Tf*Cut (orange, PDB ID: 7XTV), LCC-ICCG (blue, PDB ID: 7VVE), *Tf*Cut-DM (green, PDB ID: 7XTT), and *Is*PETase (pink, PDB ID: 7XTW).

obtained crystal structures of the *apo*-form and substrate analog complex of *Tf*Cut-DM. The crystal structure of *apo*-*Tf*Cut-DM was solved to 2.20 Å resolution and it contains two polypeptide chains in an asymmetric unit (Supplementary Table 4). Superimposing the structure with the previously reported WT *Tf*Cut yields an RMSD value of 0.255 Å for all Cα carbons, indicating that the overall structure is maintained[29]. The crystal structure of *Tf*Cut-DM shows a classical α/β-hydrolase in which nine β-sheets are sandwiched by eight α-helices on both sides (Fig. 3a). The catalytic triad residues, S170, D216, and H248, are located in the loop regions between α5 and β5, α6 and β7, and α7 and β8, respectively. The DM residues, S224 and I228, are located below the catalytic triad (Fig. 3b). It is worth noting that the side-chain of W195 in *Tf*Cut-DM switches from C-type conformation in the WT *Tf*Cut structure to B-type conformation according to the assignment in the previous publication[26]. Note that W195 only exhibits B-type in the reported *Tf*Cut-DM crystal, while the corresponding residue in the previously solved *Is*PETase crystal shows A-, B-, and C-type conformations. Since the Trp conformation is determined by the two-residue located beneath the Trp, we presumed that W195 can still wobble in the *apo*-form, though alternate conformations were not captured in this crystal[32].

We further constructed the inactive variant S170A to obtain a complex structure of BTa but failed to obtain any dataset that contains additional electron density that can be modeled with the ligand. We thus turned to a BTa-like compound MHET, which is composed of TPA and ethylene glycol linked via an ester bond (Fig. 1). As a result, the structure of the S170A variant was determined in *apo*- and MHET-bound forms to 2.21 and 1.82 Å resolutions, respectively (Supplementary Table 4). The *apo*-form of S170A did not have an observable difference compared to *Tf*Cut-DM (Cα RMSD, 0.277 Å). In the MHET-bound complex, the omit map clearly indicates the location of MHET in the active site (Supplementary Fig. 8). Notably, the ethylene glycol portion of the bound MHET was located outwards and away from the

catalytic center, opposing the direction of those in other MHET complexes (see below). Nevertheless, the hydroxyl group of S170 is estimated at 2.9 Å from the carbonyl carbon of MHET (Fig. 3c), which is within range for nucleophilic attack. Furthermore, a docking assay also revealed the same binding pose in the MHET-locating region (see below). Therefore, the complex structure should present the substrate-binding mode of *Tf*Cut-DM. The residues that ineract with MHET, Y100, M171, W195, and I218, are conserved in *Is*PETase and LCC-ICCG, while I218 has an alternative residue in LCC-ICCG (Fig. 3c and Supplementary Fig. 9).

Next, the MHET-binding pose of *Tf*Cut-DM and other PET-hydrolyzing enzymes were compared. We previously solved the crystal structure of *Is*PETase in complex with an MHET analog HEMT (Supplementary Fig. 10), which is hypothesized to mimic the substrate-binding mode[26]. To obtain a better reference, we solved the binary complex structure of the MHET-bound inactive variant of *Is*PETase (R132G/S160A) to a resolution of 1.91 Å (Supplementary Table 4) and found that MHET shares an identical binding mode to HEMT[26]. Superimposing the MHET-bound complex of *Is*PETase and *Tf*Cut-DM indicates that the ligand binding modes are similar. In both complexes, the TPA residue of MHET is bound to the TPA-interacting Trp (W195 in *Tf*Cut-DM) that adopts B-type conformation via a T-stacking force (Fig. 3d). The same type of interaction between MHET and the equivalent Trp was observed in LCC-ICCG complex. The complex structure has the conventional C-type Trp and exhibits a steeper angle relative to the ethylene moiety because its TPA residue is expelled by the hydrophobic force of the Trp indole (Fig. 3d)[26].

### Comparison of the substrate-binding mode in WT *Tf*Cut and *Tf*Cut-DM

Although the MHET complexes are based on two hydrolases that share a sequence identity as high as 51.1%, the possibility that subtle variations in the active site environment lead to the observed MHET conformation change cannot be excluded. To better illustrate whether DM strategy alters the substrate-binding mode, we solved crystal structures of an inactive WT *Tf*Cut variant (S170A) and its complex with the PBAT analog. Though the attempt to obtain BTa complex failed, we eventually succeeded in solving the MHET-bound WT *Tf*Cut complex structure (Supplementary Table 4). The overall complex structure is almost identical to *Tf*Cut-DM, although they were crystalized into different space groups (monoclinic *vs*. trigonal), and a complete density map of MHET was observed in the active site cleft of the complex structure (Supplementary Fig. 8).

The MHET complex structures of *Is*PETase, WT *Tf*Cut, *Tf*Cut-DM, together with the recently reported LCC variant (LCC-ICCG)[25] are compared in parallel (Fig. 4a). The MHET-binding-modes in WT *Tf*Cut and LCC-ICCG that harbor a TPA-binding Trp in C-type conformation are consistent, while those in *Tf*Cut-DM and *Is*PETase that harbor a TPA-binding Trp in B-type conformation are consistent (Fig. 3d). The WT *Tf*Cut contains the combination of His/Phe in positions of 224 and 228 below W195 and this pair fixes the W195-bareing loop and its indole ring in the C-type conformation. This leads to a narrower substrate entering portal (circled in Fig. 4a). The same structural feature should be shared by canonical cutinases as revealed in the LCC-ICCG complex (Fig. 4a) because the His/Phe duo is strictly conserved in similar enzymes[19]. On the other hand, the B-type Trp leads to a more open and flat TPA-binding pocket, which produces less hindrance for the entry of bulkier substrates such as PET and TPA-containing segments of PBAT (Fig. 4a). As mentioned above, the C-type W195 causes MHET to bind with a more tilted TPA compared with B-type W195. (Fig. 4b). Altogether, the increased flexibility of the substrate-binding pocket, particularly the TPA-binding Trp, should allow PBAT to be accommodated in the active site cleft in a more extended conformation.

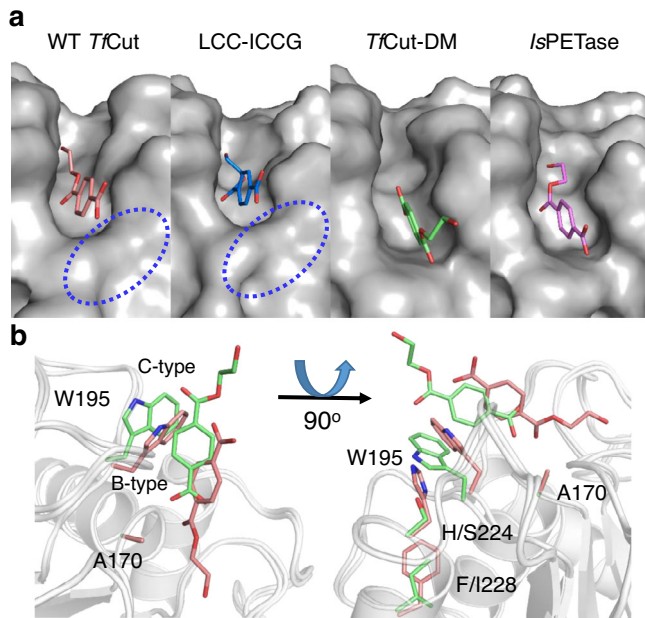

**Fig. 4 | Comparison of binding poses of MHET in WT *Tf*Cut and *Tf*Cut-DM. a** The superimposed active site clefts of MHET-bound complex of WT *Tf*Cut (PDB ID, 7XTV), LCC-ICCG (PDB ID: 7VVE), *Tf*Cut-DM (PDB ID, 7XTT), and *Is*PETase (PDB ID: 7XTW). The MHET-binding pockets are displayed with surface representation, and the dashed circles highlight the location of H224 and F228. **b** Comparison of binding modes of MHET in WT *Tf*Cut (red) and *Tf*Cut-DM (green). The selected residues are shown as sticks.

## Catalytic mechanism of *Tf*Cut-mediated PBAT hydrolysis

Compared with lipase, esterase and PBAT-hydrolytic microbes[11,30,33], our results clearly demonstrate that cutinases tend to produce TPA-terminal intermediates with BTa and TPA being the dominant products. In addition, DM variants are more efficient in yielding TPA. These results indicate that cutinases are more potent in decomposing PBAT than other types of esterase and are prone to break the ester bond adjacent to the phenyl residues. Thus, we proposed the mechanism of cutinase-mediated PBAT degradation based on the above structural and biochemical analyses.

Inspecting the structures of *Pf*L1 and Cbotu_EstA (Supplementary Table 1) shows that the catalytic center of these enzymes is deeply embedded in a pocket and is covered by a lid-domain constituted by three helices (Supplementary Fig. 11)[11,30]. The hydrophobic substrate-binding pockets show restricted space with both ends blocked (Note that the putative substrate-binding pocket of *Pf*L1 is broken into two by residues that extend from the lid-domain). Compared with enzymes whose substrate-binding tunnel is open on both ends, these two enzymes are considered more suitable to accommodate short-chain substrates instead of long-chain substrates such as PBAT. As the MHET observed in the *Tf*Cut-DM complex structure shows a binding pose with its ethylene glycol pointing outside the enzyme, we conducted a docking assay to simulate the substrate-binding mode within the enzyme. A representative model of PBAT, BABTaB, was chosen for the docking assay (Fig. 5a). As expected, the TPA moiety aligns well with that in the MHET complex structure. The aliphatic parts stretch to fill a cleft across the protein surface (Fig. 5a). This tunnel is lined by several amino acids, including H169, W195, K199, H248, N252, G99, Y100, I218, and T223. Together with the biochemical and crystallographic analyses, we proposed the *Tf*Cut-catalyzed degradation of PBAT. As shown in Fig. 5b, *Tf*Cut recognizes the substrate by locating the TPA moiety in the hydrophobic substrate-binding pocket, including the TPA-binding W195. This pose facilitates the catalytic residue S170 to attack the ester

bond adjacent to the TPA-terminus. In the initial stage of reaction, *Tf*Cut-mediated hydrolysis should practice an 'endo' fashion, which causes random cleavage of the macromolecules. When smaller hydrolytic intermediates are generated, exo-reaction in addition to endo-reaction can occur, which might catalyze hydrolysis toward substrates harboring TPA on at least one end. Further digestion enables *Tf*Cut to turn intermediates into TPA and BTa. As for *Tf*Cut-DM, its higher TPA-binding capacity should contribute to the complete hydrolysis of all intermediates to TPA on the same time scale.

The bio-degradation of polyesters such as PBAT and PET has attracted much attention as this environmental-friendly approach provides a solution to manage the problem of plastic pollution. Furthermore, decomposing PBAT to release monomeric components that can serve as virgin-like materials for synthesizing the same polymers or be converted to other materials for open-loop recycling and upcycling is also an important direction for the sustainable utilization of PBAT[3,34]. However, the catalytic rate of depolymerases is still not efficient enough for further applications. In this study, we demonstrate that cutinases as well as PET-hydrolytic enzymes are more effective than other types of esterase. Among them, two thermophilic cutinases that can completely degrade PBAT film in 48 h possess further application potentials. Moreover, engineering cutinases via the DM strategy enhanced their hydrolytic efficacy towards PBAT, shortened the time required for complete PBAT film decomposition, and increased the yield of TPA (Fig. 2d and Supplementary Fig. 6a). The DM variants also show advantages over the wild type enzymes in degrading UV-irradiated films. The structural analyses indicate that the various substrate-binding modes in WT and DM enzymes could result from the conformation change of the TPA-binding Trp. The flexible substrate-binding pocket of DM variants may be the key feature that leads to the enhanced hydrolysis efficiency towards PBAT as well as PET (Fig. 4a). Since TPA is a fossil fuel-derived feedstock for polymer production, DM variants that hydrolyze PBAT and yield TPA as the dominant product should possess higher application potential in the plastic circular economy[1,35]. These findings provide the molecular understanding of effective enzyme-mediated PBAT-degradation, which will facilitate further enzyme engineering to advance the development of PBAT biodegradation in the future.

## Methods
### Chemicals
The kits for crystallization screening were purchased from Hampton Research (Aliso Viejo, USA). The crystallization reagents, including lithium sulfate, sodium acetate, polyethylene glycol (PEG) 8000, imidazole malate, sodium cacodylate, and ammonium sulfate, were purchased from Sigma-Aldrich (St. Louis, USA). The polymer PBAT films were purchased from Shanghai Hongrui Biotechnology Co., Ltd (Shanghai, CHINA). The standard compounds TPA and BTa were purchased from J&K Scientific Ltd. (Beijing, CHINA) and Bide Pharmatech Ltd. (Shanghai, CHINA), respectively. Based on ¹H NMR spectra analysis[17,31], the Ada (adipic acid)/TPA ratio of the film is estimated to be 1 (Supplementary Fig. 12), which is identical to Ecoflex® that is manufactured by BASF (Ludwigshafen, Germany)[31].

### Gene cloning and protein purification
The recombinant plasmids of pET-32a carrying coding genes of WT and DM variant of *Tf*Cut, *Is*PETase, *Pb*PL, and *Bur*PL were constructed into the plasmids such as pET32a[19]. The gene encoding *Tc*Cut from *Thermobifida cellulosilytica* (GenBank accession number: ADV92526.1) was chemically synthesized by Sangon Biotech (Wuhan, China) and cloned to pET-32a. The *Tc*Cut-DM plasmid was constructed by site-directed mutagenesis using primers: 5′- ATTGCACCGGTTGCGACCT CTGCAAAACCGATTTATAACAGCCTGCCGAGC−3′. The primers for constructing the S170A variant of WT *Tf*Cut and *Tf*Cut-DM are 5′-GGC

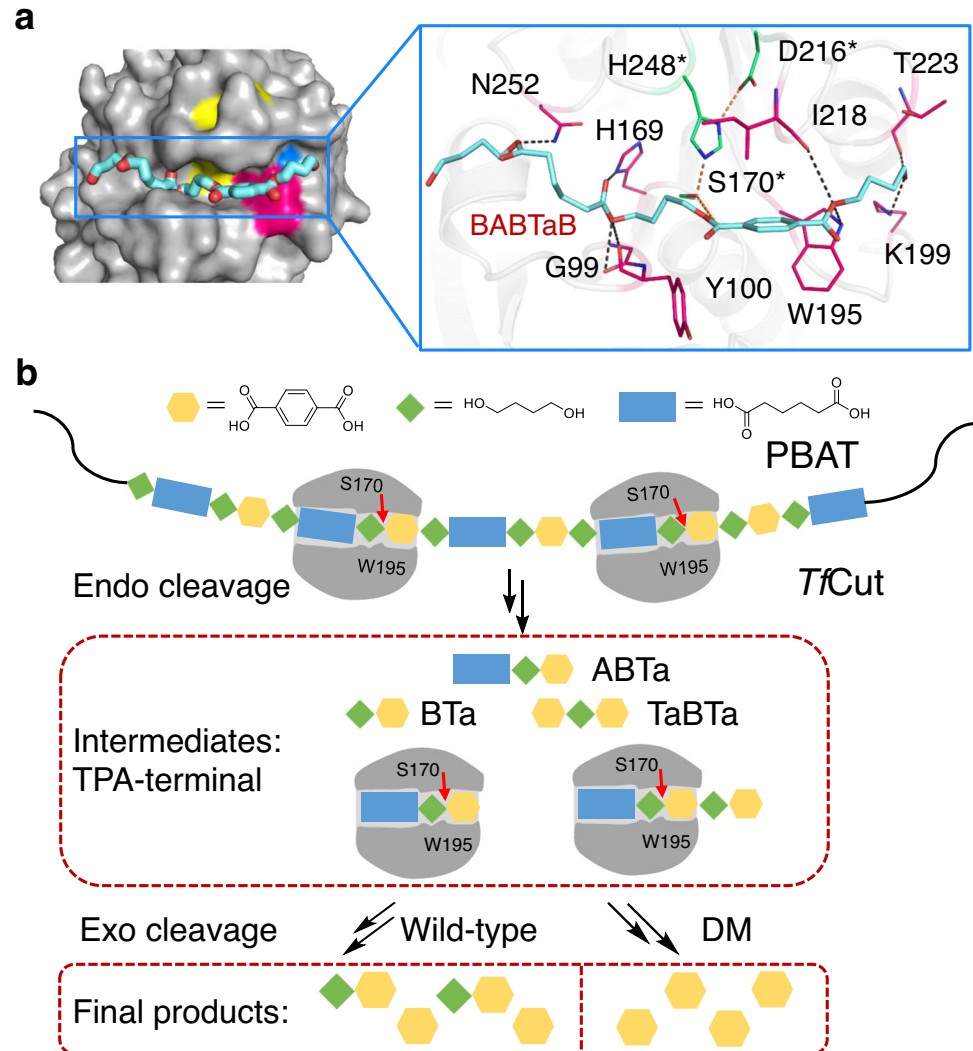

**Fig. 5 | *Tf*Cut-catalyzed PBAT degradation. a** The predicted binding mode of the polymer chain of PBAT in the active site tunnel of *Tf*Cut-DM. The polymer chain is drawn as BABTaB (cyan sticks). The catalytic triad, W195, and DM are colored in yellow, magenta, and blue, respectively. The zoom in figure is shown on the right side. The interaction network of BABTaB with surrounding residues in *Tf*Cut-DM is shown in magenta sticks and gray dash lines (<3.5 Å). The catalytic triad residues are shown in green sticks and the interactions are shown with orange dash lines. **b** The proposed method of *Tf*Cut-mediated PBAT degradation, including endo- and exo-cleavage. The representative PBAT polymer is drawn as the repeats of BABTaB and other combinations. The major intermediates that have TPA on the termini are depicted.

AGTTATGGGTCATGCAATGGGTGGTGGTGGTAGCC-3' and 5'- ATTGCA CCGGTTGCGACCTCTGCAAAACCGATTTATAACAGCCTGCCGAGC-3'.

All recombinant proteins of the abovementioned plasmids were expressed and purified following the same procedures as previously reported[19,26]. The recombinant plasmid was transformed into *Escherichia coli* BL21(DE3) cells that were grown in LB medium at 37 °C to an optical density at 600 nm of ca. 0.8. The protein expression was induced by 0.5 mM isopropyl β-D-thiogalactopyranoside at 16 °C for 20 h. Cells were collected by centrifugation at 6000 × *g* for 10 min and then resuspended in a lysis buffer containing 20 mM Tris-HCl, 300 mM NaCl and 20 mM imidazole (pH 8.0), followed by disruption with a French press. Cell debris was removed by centrifugation at 27,000 × *g* for 1 h. The supernatant was then applied to a Ni-NTA column with a fast protein liquid chromatography (FPLC) system (GE Healthcare). The target protein was eluted using an imidazole gradient from 20 mM to 250 mM and the protein-containing fractions were collected. The protein solution was dialyzed against a buffer containing 20 mM Tris-HCl and 300 mM NaCl (pH 8.0) and subjected to the tobacco etch virus protease digestion overnight to remove the 6× His-tag on the N terminal. The mixture was then passed through a Ni-NTA column one more time to remove the His-tagged portion, with the untagged protein eluted in the imidazole-free buffer. The protein purity (> 95%) was verified by SDS–PAGE analysis.

**Identification and quantitation of products in PBAT degradation**
The optimal reaction buffer for all recombinant enzymes except *Tc*Cut is 50 mM glycine-NaOH buffer (pH 9.0) according to the previous report[19]. The optimal reaction buffer of *Tc*Cut was determined by incubating the recombinant enzyme with various buffers and a piece of PBAT film (1 × 1 cm) at 60 °C for 12 h, and the hydrolytic products were analyzed by the method described below. The pH profile of *Tc*Cut is shown in Supplementary Fig. 13a, b, which indicates the optimal buffer for *Tc*Cut is also 50 mM glycine-NaOH buffer (pH 9.0). The optimal operating temperatures of WT *Tc*Cut and *Tc*Cut-DM were subsequently determined to 65 and 60 °C, respectively (Supplementary Fig. 13c). For all reactions, 10 μg/mL enzyme was incubated in 50 mM glycine-NaOH buffer (pH 9.0) and a piece of PBAT film (1 × 1 cm) at indicated temperatures with agitation at 800 rpm for the indicated period of time. The reactions were terminated by boiling for 10 min, passed through a 0.22 μm filter, and centrifuged at 12,000 rpm for 10 min before being

applied to a high-performance liquid chromatography system (HPLC, Shimadzu LC-20AD, JAPAN) equipped with an InerSustain $C_{18}$ column ($4.6 \times 250$ mM, 5 μm). The $C_{18}$ column was eluted with solvent A (0.1% formic acid) and solvent B (acetonitrile) with a linear gradient of 25% to 85% methanol in 35 min. The flow rate of the mobile phase was 1.0 mL/min, and the effluent was monitored at a wavelength of 254 nm. The data of mass spectra (MS) was collected in a negative mode by the instrument of Thermos TSQ Quantum Ultra (Waltham, USA).

The amounts of TPA and BTa were calculated based on the calibration functions from their standard samples. The enzyme activity was calculated by the sum of released BTa and TPA. All samples were analyzed in triplicate independently in each experiment.

### Crystallization and structure determination
All protein crystallization was conducted at 25 °C using the sitting-drop vapor-diffusion method. In general, 1 μL of protein-containing solution (20–30 mg/mL) was mixed with 1 μL reservoir solution in 96-well Cryschem plates. The crystallization conditions for each protein are as follows: WT *Tf*Cut S170A, 4% *v/v* PEG 200, 0.8 M lithium sulfate and 0.1 M sodium acetate, pH 4.0; *Tf*Cut-DM, 2% *w/v* PEG 8000, 1.0 M lithium sulfate and 0.1 M imidazole malate, pH 7.5; *Tf*Cut-DM S170A, 30% *w/v* PEG 8000, 0.2 M ammonium sulfate and 0.1 M sodium cacodylate, pH 6.5; *Is*PETase R132G/S160A, 1.6 M ammonium sulfate, 0.1 M Tris-HCl, pH 8.0. To obtain the complex structure, the crystals were soaked with the mother liquid containing 15 mM MHET, which was made by the procedure described previously[19].

Before the data collection, all crystals were soaked with cryoprotectants (the mother liquid contains 10-20% ethylene glycol or glycerol). The X-ray diffraction datasets were collected by Bruker D8 Venture coupled with a CMOS-PHOTON III detector at Hubei University. The program PROTEUM 3 was used to process the X-ray diffraction datasets. The structures were solved by the method of molecular replacement (MR) with the *Tf*Cut *apo* structure (PDB ID, 4CG3) as a search model. Subsequent model adjustment and refinement were conducted by using Refmac5, PHENIX, and Coot[36–38]. Prior to structure refinement, 5% of randomly selected reflections were set aside for calculating $R_{\text{free}}$ as a monitor of model quality. All protein structure figures were prepared using the PyMOL program (http://pymol.sourceforge.net/).

### Docking the model of PBAT into *Tf*Cut-DM
The docking experiment was performed with LeDock software[39], which employs a combination of simulated annealing and genetic algorithm to optimize the binding position and orientation of the docked molecules. The reference structure is MHET-bound *Tf*Cut-DM (PDB ID: 7XTT), in which waters and MHET were removed and residues were protonated in the module of LePro. The representative model of PBAT, BABTaB, was input into the module of LeDock with a Mol2 format. The position of TPA moiety in MHET-bound *Tf*Cut-DM was set as the center point. The size of the grid box was set as ca. $10 \times 10 \times 30$ Å$^3$ to make sure to embed the active site tunnel of *Tf*Cut-DM. The outcome contains various binding poses in descending order and the top one was selected for further analysis.

### Gel content of UV-irradiated PBAT film
The PBAT film was first exposed to a UV lamp (PHILIPS 8 W) for 24 h on each side and dissolved in 20 mL CHCl₃ for 48 h. Based on the method described by Hoe et al.[31], the gel content of the UV-irradiated PBAT film was calculated by measuring the ratio of the insoluble portion after drying in a hood to the initial weight. The calculated gel content is estimated as 55% after UV exposure.

### Reporting summary
Further information on research design is available in the Nature Portfolio Reporting Summary linked to this article.

## Data availability
The atomic coordinates and structure factors have been deposited in the Protein Data Bank for reported structures of the Native *Tf*Cut-DM (PDB ID, 7XTR), *Tf*Cut-DM S170A (PDB ID, 7XTS), MHET-bound *Tf*Cut-DM S170A (PDB ID, 7XTT), WT *Tf*Cut S170A (PDB ID, 7XTU), MHET-bound WT *Tf*Cut S170A (PDB ID, 7XTV), and MHET-bound *Is*PETase S160A (PDB ID, 7XTW). The source data are provided as a Source Data file. Source data are provided with this paper.

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

## Acknowledgements
This work was supported by the National Key Research and Development Program of China (2021YFC2104000, 2022YFE0135300, 2019YFA0706900, and 2018YFE0204503), Hubei Hongshan Laboratory (2022hszd030), the National Natural Science Foundation of China (32271318, 32101016, and 81871251); the Natural Science Foundation of Hubei Province (2020CFA011, 2022CFA101, and 2022CFB360).

## Author contributions
R.T.G., Y.L., and C.C.C. designed and supervised researches. Y.Y., X.T., P.J., Y.Q., and X. Li. carried out cloning, mutagenesis, protein purification, and crystallization. Y.Y., J.W.H., N.M., F.C.T., Q.Z., Y.L., C.C.C., and R.T.G. analyzed the crystallographic data. Y.Y., J.M., X.T., P.J., X. Liu, R.P., and L.D. measured the hydrolytic activity and analyzed the data. Y.Y., C.C.C., and R.T.G. wrote the manuscript with contributions from all authors.

## Competing interests
The authors declare no competing interests.
