## [Peer Review File · Nature Communications]

Complete bio-degradation of poly(butylene adipate-co-terephthalate) via engineered cutinasesEditorial Note: This manuscript has been previously reviewed at another journal that is not operating a transparent peer review scheme. This document only contains reviewer comments and rebuttal letters for versions considered at *Nature Communications*.

REVIEWERS' COMMENTS

Reviewer #1 (Remarks to the Author):

The authors report on the bio-degradation of the polymer PBAT by a cutinase from *Thermobifida fusca* (TfCut). For the original submission, I was asked to focus my review on the structural biology aspects of the work, and I raised a couple of points regarding the presentation of the results and the details of the docking calculation.

In the transferred and revised version, the authors adequately addressed all my concerns and corrected all issues.

Reviewer #2 (Remarks to the Author):

The article by Yu Yang et al demonstrates how several thermophilic cutinases, in particular those from *Thermobifida cellulositica* and *Thermobifida fusca*, can be used as biocatalysts for the depolymerization of the aliphatic-aromatic copolymer PBAT into several intermediates, including the terephthalate moiety that is used for synthesizing virgin polymers. Moreover, they demonstrate that a previously described double substitution based on a mesophilic PET-degrading enzyme (IsPETase), which enables an active site tryptophan to rotate into different catalytically competent conformations and improves the activity of several cutinases against PET (doi: 10.1038/s41929-021-00616-y), enables a cutinase from *Thermobifida fusca* (TfCut) to efficiently degrade PBAT into TPA as the major degradation product (98% conversion).

The manuscript is a strongly improved version of a previous version submitted to another Nature journal, which I had the pleasure of being a reviewer (unfortunately as reviewer #2!). In this regard, the authors present new results to demonstrate which of the cutinases are more efficient to degrade PBAT, including the detailed characterization of previously reported enzymes that were known to depolymerize PBAT (such as the one from *T. cellulositica*) for a better comparison against their proposed best enzyme TfCut. The authors also improved their manuscript by considering the optimal temperatures for PBAT degradation of all enzymes in their comparisons, as well as proposing a mechanism for the hydrolysis of PBAT based on the crystal structures that they obtained in the absence and presence of a PET degradation product (MHET) that is quite similar to one of the PBAT degradation products determined by HPLC and mass spectrometry during the depolymerization reactions.

The results are noteworthy and sound, as they not only report on the activities of the wild-type and doubly-substituted variants for several cutinases and their optimal temperatures for degradation, but also test the activity against UV-irradiated PBAT films that best represent the post-consumer conditions of this co-polymer, which could be a relevant aspect when considering these enzymes as potential biocatalysts for use in industrial setups. The manuscript also contains sufficient methodological details to reproduce these experiments and further improve these enzymes through the myriad techniques of artificial intelligence and directed evolution that are being developed by the scientific community, with the results being highly relevant for the several groups working on the potential biological recycling of plastics worldwide.

My minor comments are as follows:

1) It would be great if you can indicate the protein purification yields for the different enzymes, as their production yield is also a relevant aspect for their potential further use in industry.

2) Regarding the affirmation in lines 200-202 that, for TfCut-DM, "we presumed that W195 can still wobble in apo-form status though an alternative conformation was not captured in this crystal", I would recommend the authors to cite recent literature, if possible, that suggest that these alternative conformations are in fact accesible for the wild-type or doubly-substituted mutants of these cutinases.

3) There are some sentences that might need slight improvements. For example:

a) Lines 50-60: "PBAT was firstly considered as a compostable biopolymer considering..."; you can possible rephrase this

b) Lines 66-68: "Applying bio-based degradation using renewable biological entities, including enzymes or microorganisms, is an ideal and environmentally benign approach to reduce even recycle plastics"; I believe you mean "and even recycle plastics"?

c) Lines 77-68: "Nevertheless, the PBAT decomposition rates of lipase or esterase are generally low."; perhaps it should be "lipases" and "esterases" in plural.

Reviewer #3 (Remarks to the Author):

The authors have thoughtfully responded to all the comments raised by the reviewers and addressed concerns through additional experiments. Notable is the expanded experimental activity PBAT hydrolytic enzyme library presented in the Supplementary Table 1 and the finding of TcCut activity which was not previously presented. This manuscript is much improved, and I support publication following minor changes to Extended Data Figure 8.

1. Line 218 does not match those presented in Extended Data Figure 8 which causes confusion to the reader. Please change the numbers on the figure to correspond to the full-length sequence and/or indicate in the text the numbering discrepancy.

2. Additionally, remove the purple star corresponding to R46 per your editorial response.

REVIEWERS' COMMENTS

Reviewer #1 (Remarks to the Author):

The authors report on the bio-degradation of the polymer PBAT by a cutinase from *Thermobifida fusca* (TfCut). For the original submission, I was asked to focus my review on the structural biology aspects of the work, and I raised a couple of points regarding the presentation of the results and the details of the docking calculation.

In the transferred and revised version, the authors adequately addressed all my concerns and corrected all issues.

Response: We thank the Reviewer for the positive comments.

Reviewer #2 (Remarks to the Author):

The article by Yu Yang et al demonstrates how several thermophilic cutinases, in particular those from *Thermobifida cellulositica* and *Thermobifida fusca*, can be used as biocatalysts for the depolymerization of the aliphatic-aromatic copolymer PBAT into several intermediates, including the terephthalate moiety that is used for synthesizing virgin polymers. Moreover, they demonstrate that a previously described double substitution based on a mesophilic PET-degrading enzyme (IsPETase), which enables an active site tryptophan to rotate into different catalytically competent conformations and improves the activity of several cutinases against PET (doi: 10.1038/s41929-021-00616-y), enables a cutinase from *Thermobifida fusca* (TfCut) to efficiently degrade PBAT into TPA as the major degradation product (98% conversion).

The manuscript is a strongly improved version of a previous version submitted to another Nature journal, which I had the pleasure of being a reviewer (unfortunately as reviewer #2!). In this regard, the authors present new results to demonstrate which of the cutinases are more efficient to degrade PBAT, including the detailed characterization of previously reported enzymes that were known to depolymerize PBAT (such as the one from *T. cellulositica*) for a better comparison against their proposed best enzyme TfCut. The authors also improved their manuscript by considering the optimal temperatures for PBAT degradation of all enzymes in their comparisons, as well as proposing a mechanism for the hydrolysis of PBAT based on the crystal structures that they obtained in the absence and presence of a PET degradation product (MHET) that is quite similar to one of the PBAT degradation products determined by HPLC and mass spectrometry during the depolymerization reactions.

The results are noteworthy and sound, as they not only report on the activities of the wild-type and doubly-substituted variants for several cutinases and their optimal temperatures for degradation, but also test the activity against UV-irradiated PBAT

films that best represent the post-consumer conditions of this co-polymer, which could be a relevant aspect when considering these enzymes as potential biocatalysts for use in industrial setups. The manuscript also contains sufficient methodological details to reproduce these experiments and further improve these enzymes through the myriad techniques of artificial intelligence and directed evolution that are being developed by the scientific community, with the results being highly relevant for the several groups working on the potential biological recycling of plastics worldwide.

Response: We thank the Reviewer for the positive comments.

My minor comments are as follows:

1) It would be great if you can indicate the protein purification yields for the different enzymes, as their production yield is also a relevant aspect for their potential further use in industry.

Response: We thank the Reviewer for raising the question concerning the enzyme production. The production yield of the enzymes mentioned in the manuscript has been added in the **Supplementary Table 1**.

2) Regarding the affirmation in lines 200-202 that, for TfCut-DM, "we presumed that W195 can still wobble in apo-form status though an alternative conformation was not captured in this crystal", I would recommend the authors to cite recent literature, if possible, that suggest that these alternative conformations are in fact accesible for the wild-type or doubly-substituted mutants of these cutinases.

Response: We thank the Reviewer for pointing out this issue. The relative reference has been cited as #32.

Crnjar, A., Griñen, A., Kamerlin, S. C. L. & Ramírez-Sarmiento, C. A. Conformational Selection of a Tryptophan Side Chain Drives the Generalized Increase in Activity of PET Hydrolases through a Ser/Ile Double Mutation. *ACS Org. Inorg. Au* (2023).

3) There are some sentences that might need slight improvements. For example:

a) Lines 50-60: "PBAT was firstly considered as a compostable biopolymer considering..."; you can possible rephrase this

Response: We thank the Reviewer for pointing out the mistake. It has been modified to "PBAT was firstly considered as a compostable biopolymer because polyesters are more amenable to enzymes such as esterases".

b) Lines 66-68: "Applying bio-based degradation using renewable biological entities, including enzymes or microorganisms, is an ideal and environmentally benign approach to reduce even recycle plastics"; I believe you mean "and even recycle plastics"?

Response: It has been corrected.

c) Lines 77-68: "Nevertheless, the PBAT decomposition rates of lipase or esterase are

generally low."; perhaps it should be "lipases" and "esterases" in plural.

Response: It has been corrected.

Reviewer #3 (Remarks to the Author):

The authors have thoughtfully responded to all the comments raised by the reviewers and addressed concerns through additional experiments. Notable is the expanded experimental activity PBAT hydrolytic enzyme library presented in the Supplementary Table 1 and the finding of TcCut activity which was not previously presented. This manuscript is much improved, and I support publication following minor changes to Extended Data Figure 8.

Response: We thank the Reviewer for the positive comments.

1. Line 218 does not match those presented in Extended Data Figure 8 which causes confusion to the reader. Please change the numbers on the figure to correspond to the full-length sequence and/or indicate in the text the numbering discrepancy.

2. Additionally, remove the purple star corresponding to R46 per your editorial response.

Response: We thank the Reviewer for pointing out the mistake. The **Extended Data Figure 8** (now is the **Supplementary Figure 9**), has been replaced with an updated numbers of residues in *TfCut*. The purple star under R46 has been removed as well.